# Research on Fault Diagnosis of Rolling Bearing Based on Gramian Angular Field and Lightweight Model

**DOI:** 10.3390/s24185952

**Published:** 2024-09-13

**Authors:** Jingtao Shen, Zhe Wu, Yachao Cao, Qiang Zhang, Yanping Cui

**Affiliations:** 1School of Mechanical Engineering, Hebei University of Science and Technology, Shijiazhuang 050018, China; shenjingtaohblg@163.com (J.S.); yachao.cao@hebust.edu (Y.C.); cuiypkd@163.com (Y.C.); 2Key Laboratory of Vehicle Transmission, China North Vehicle Research Institute, Beijing 100072, China; qiangzh36@gmail.com

**Keywords:** deep learning, fault diagnosis, residual network, Gramian angular field, efficient channel attention

## Abstract

Due to the limitations of deep learning models in processing one-dimensional signal feature extraction, and high model complexity leading to low training accuracy and large consumption of computing resources, this paper innovatively proposes a rolling bearing fault diagnosis method based on Gramian Angular Field (GAF) and enhanced lightweight residual network. Firstly, the one-dimensional signal is transformed into a two-dimensional GAF image, fully preserving the signal’s temporal dependency. Secondly, to address the parameter redundancy and high computational complexity of the ResNet-18 model, its residual blocks are improved. The second convolutional layer in the downsampling residual blocks is removed, traditional convolutional layers are replaced with depthwise separable convolutions, and the lightweight Efficient Channel Attention (ECA) module is embedded after each residual block. This further enhances the model’s ability to capture key features while maintaining low computational cost, resulting in a lightweight model referred to as E-ResNet13. Finally, the generated GAF feature maps are fed into the E-ResNet13 model for training, and through a global average pooling layer, they are mapped to a fully connected layer for classifying the faults of rolling bearings. Verifying the superiority of the proposed GAF-E-ResNet13 model, experimental results show that the GAF image encoding method achieves higher fault recognition accuracy compared to other encoding methods. Compared with other intelligent diagnosis methods, the E-ResNet13 model demonstrates strong diagnostic performance and generalization capability under both a single condition and complex varying conditions, fully proving the innovation and practicality of this method.

## 1. Introduction

With the development of industrialization in China, the application of rotating machinery in large-scale equipment has become increasingly widespread. Rolling bearings play a crucial role in the operation of such machinery. However, prolonged use of rolling bearings can lead to wear and failure, causing machinery to be prone to malfunctions and, in severe cases, resulting in injury or even fatalities to personnel. According to statistics, more than 50% of mechanical failures are related to bearings [1,2]. Therefore, research on intelligent algorithms for bearing fault diagnosis is crucial to ensure the normal operation of mechanical equipment.

The data generated by mechanical equipment exhibit characteristics such as multiple operating conditions, severe coupling of fault information, and significant environmental noise pollution [3,4]. To address these issues, numerous scholars have proposed traditional machine-learning-based intelligent diagnostic algorithms, including Artificial Neural Networks (ANN) [5], Support Vector Machines (SVM) [6,7], Random Forests [8], and K-Nearest Neighbors (KNN) [9]. However, traditional machine-learning-based fault diagnosis methods still have certain limitations: (1) Machine learning methods require extensive data processing for diagnostic tasks, heavily relying on rich experience and prior knowledge of expert systems. (2) The structure of machine learning models is simple, with shallow layers, unable to extract deeper features, resulting in low fault diagnosis accuracy. (3) During actual signal acquisition, there is often a large amount of noise, making it difficult for simple machine learning methods to handle nonlinear signals, and limiting their feature extraction capabilities.

Based on the above issues, Hinton et al. [10] introduced the concept of deep learning, where Convolutional Neural Networks (CNNs) represent one of the most significant algorithms. CNNs can automatically extract features from raw signals, avoiding the need for extensive data labeling, and have gradually been applied in the field of bearing fault diagnosis. CNN models utilize convolutional kernels of various sizes to extract features from bearing fault signals, thereby enhancing diagnostic effectiveness by enlarging the receptive field of convolution operations [11,12]. Deep learning is not only applied in fault diagnosis but also involved in the detection of abnormal machine sounds; for example, Yan et al. [13] proposed an unsupervised machine abnormal sound detection model based on transformer and dynamic graph convolution (Unsuper-TDGCN). The method has been validated on multiple datasets and has achieved promising results. With continuous innovations in CNN models, many researchers have leveraged the advantages of CNN by increasing the complexity of their structures to extract deeper features. However, deeper models introduce several challenges: (1) Gradient vanishing and explosion: In deep networks, gradients can become extremely small or large as they propagate through layers, leading to unstable weight updates and affecting training effectiveness. (2) Increased computational resource consumption: Deeper network structures require more computation and memory resources, which not only prolongs training times but also escalates hardware resource requirements. (3) Increased risk of overfitting: Complex network structures may perform well on training data but poorly on testing data, reducing generalization capability and practical application effectiveness, potentially causing model breakdown.

Based on the above analysis, He et al. [14] proposed the Deep Residual Network (ResNet) to address issues arising from increasing network depth through a special structure called “shortcut connections”. Consequently, ResNet models have been widely applied in fault diagnosis and have achieved excellent diagnostic performance. Studies [15,16] have combined ResNet models with attention mechanisms, not only deepening the network structure but also enhancing the efficiency of extracting useful features. Wen et al. [17] proposed a deep learning model named TCNN, based on the ResNet-50 model and transfer learning. The model demonstrated high testing accuracy across different fault datasets. Meng et al. [18] proposed a diaphragm pump fault diagnosis model called PCA-ResNet. Compared to other models, it exhibits higher operational efficiency and lower loss rates, effectively addressing the issue of diagnostic lag in practical engineering applications.

However, as deep learning model structures become more complex, one-dimensional signals cannot fully reflect spatial relationships between data points, making it difficult for models to comprehensively capture complex information. To overcome these limitations, researchers have proposed various image encoding methods, such as Gramian Angular Field (GAF) [19], Markov Transition Field (MTF) [20], Continuous Wavelet Transform (CWT) [21], and Recurrence Plots (RP) [22]. These image encoding methods enhance the feature representation of signals, achieving higher fault diagnosis accuracy and efficiency. Studies [23,24] combining GAF and MTF image encoding methods with ResNet have demonstrated excellent diagnostic performance in bearing experiments, fully validating the advantages of a two-dimensional feature image. Xu et al. [25] combined Continuous Wavelet Transform (CWT) time-frequency images with a residual network and Support Vector Machine (SVM), achieving an accuracy of 97.54% in fault diagnosis applications. Chung et al. [26] converted one-dimensional signals into Recurrence Plot (RP) images and combined them with two deep learning models, achieving promising results in train wheel fault diagnosis.

Different encoding methods can transform signals from various perspectives and dimensions, capturing varied feature information. Studies [27,28] have shown that the fusion of GAF and MTF can provide richer feature information, improving the model’s ability to identify fault features, thus enhancing classification accuracy. With the adoption of image encoding methods, lightweight design of deep learning models has also become increasingly popular. The concept of lightweight models originates from the need for models with lower memory requirements and higher prediction efficiency in practical applications. Lightweight models reduce the number of network parameters through methods like convolution kernel decomposition and singular value decomposition, thereby accelerating network computation speed [29]. Studies [30,31] combining GAF image encoding methods with lightweight models have demonstrated excellent performance in fault diagnosis. Compared to traditional models, lightweight models not only have fewer parameters and computations but also exhibit higher fault recognition accuracy and better generalization performance. Wang et al. [32] proposed a lightweight improved residual network called LIResNet, which demonstrated high diagnostic capability and lightweight performance on bearing datasets. Dou et al. [33] proposed a gearbox fault diagnosis method based on Gramian Angular Field (GAF) and a lightweight Convolutional Neural Network. Experimental validation showed that this network can achieve fault diagnosis with limited samples under various operating conditions. Meng et al. [34] proposed a lightweight model for scale feature modules, which demonstrated a high fault recognition rate across two rolling bearing datasets. Wu et al. [35] enhanced the model’s generalization ability by combining a deep adaptation network with a lightweight model, and the feasibility of the approach was demonstrated through experimental results. Lian et al. [36] constructed a lightweight feature enhancement network using images generated by a cross-domain image fusion module as input, achieving an average accuracy of 100% in experiments. Gu et al. [37] transformed one-dimensional signals into recurrence plots and input them into the lightweight model MobileNet-v3 for training. The results demonstrated that the proposed RP + MobileNet-v3 model achieved higher diagnostic accuracy with fewer parameters. Tang et al. [38] proposed a novel lightweight network using the Ghost module, and its effectiveness was validated through three bearing datasets. Based on these research results, image encoding methods and lightweight design have gradually become important development directions for deep learning models in practical industrial applications.

Based on the above analysis, this paper proposes a lightweight improvement to the residual blocks in ResNet-18 by integrating the ECA, resulting in a network model named E-ResNet13. E-ResNet13 significantly reduces the number of parameters while also shortening training time to a certain extent, thereby effectively reducing the demand for hardware resources. By using the GAF image encoding method, one-dimensional signals are converted into two-dimensional feature images, which are then fed into the E-ResNet13 model. Extensive experiments have fully validated the effectiveness and generalization of this method. The main contributions are as follows:(1)To address the limitations of one-dimensional signals in feature representation and their insufficiencies in temporal dependency, the GAF encoding method is used to transform one-dimensional time series signals into two-dimensional images. GAF generates an information-dense image that simultaneously preserves the amplitude information of the original signal and its temporal dependency by converting each pair of data points in the time series into angles and computing their sine and cosine’s differences and sums. This transformation not only significantly enhances the separability of features but also allows the network model to more comprehensively capture and analyze complex fault characteristics while preserving temporal information, thereby improving the accuracy and reliability of diagnostics.(2)To address the issue of the large number of parameters in deep learning models, we propose a lightweight E-ResNet13 model. Firstly, some convolutional layers in the residual blocks are removed to reduce redundant computations. Secondly, traditional convolutions are replaced with depthwise separable convolutions, decreasing the number of model parameters and computations. Finally, the ECA lightweight module is integrated to adaptively reallocate weights across different channels, further enhancing the model’s focus on critical information and the accuracy of feature extraction.(3)The GAF-E-ResNet13 method demonstrates superiority and generalizability. We conducted experiments using the proposed method on the CWRU rolling bearing fault dataset under both a single condition and complex varying conditions. Compared with other intelligent algorithms, the effectiveness and practicality of the GAF-E-ResNet13 method were successfully validated.

## 2. Theory

### 2.1. Gramian Angular Field

The Gramian Angular Field (GAF) is a method that transforms one-dimensional time series into two-dimensional images, preserving the temporal dependencies and correlations of the original signals, thereby enhancing inter-signal relationships. The GAF method includes the Gramian Angular Difference Field (GADF) and the Gramian Angular Summation Field (GASF). It converts signals into polar coordinates and encodes them using a Gramian matrix where each element represents a trigonometric function of the angles between different points in time. The specific steps are as follows:

Step 1: Assume a time series signal $X=\left\{x_{1}\right.,x_{2},\cdots ,\left.x_{n}\right\}$. Normalize all values in *X* to the interval [−1, 1] or [0, 1] using the following formula [39]:(1)$${\tilde{x}}_{i}=\frac{\left(x_{i}-max\left(X\right)\right)+\left(x_{i}-min\left(X\right)\right)}{max\left(X\right)-min\left(X\right)}$$
(2)$${\tilde{x}}_{i}=\frac{x_{i}-min\left(X\right)}{max\left(X\right)-min\left(X\right)}$$

Step 2: Represent the scaled time series X in polar coordinates [40]:(3)∅=arccosx˜i,−1≤x˜i≤1,x˜i∈X˜r=ti/N,ti∈ℕ
where ti is the timestamp, N is the total number of time points in the time series data, and ∅ is the polar coordinate of the angular cosine. Through polar coordinate transformation, time information and numerical information are mapped onto the same dimension. According to the formula for vector dot product, the dot product of two time points can be approximated as the cosine of the sum of their transformed angular angles, as shown in Formula (4) [41]:(4)x1,x2=cos∅1+∅2
where x1,x2 represents the inner product of two sequential points *x*_1_ and *x*_2_, and cos(∅1+∅2) represents the cosine value of the sum of two angles.

Step 3: Construct the Gramian matrix using the transformed data. Use the values of the sum or difference of angles between each pair of points to reflect their correlation. Formula (4) [42] calculates the cosine of the sum of angles, representing GASF; Formula (5) [43] calculates the sine of the difference of angles, representing GADF.
(5)$$F_{GAS}=\left[\begin{array}{cccc} cos\left(\emptyset_{1}+\emptyset_{1}\right) & cos\left(\emptyset_{1}+\emptyset_{2}\right) & \cdots & cos\left(\emptyset_{1}+\emptyset_{n}\right)\\ cos\left(\emptyset_{2}+\emptyset_{1}\right) & cos\left(\emptyset_{2}+\emptyset_{2}\right) & \cdots & cos\left(\emptyset_{2}+\emptyset_{n}\right)\\ \cdots & \cdots & \cdots & \cdots \\ cos\left(\emptyset_{n}+\emptyset_{1}\right) & cos\left(\emptyset_{n}+\emptyset_{2}\right) & \cdots & cos\left(\emptyset_{n}+\emptyset_{n}\right) \end{array}\right]$$
(6)$$F_{GAD}=\left[\begin{array}{cccc} sin\left(\emptyset_{1}-\emptyset_{1}\right) & sin\left(\emptyset_{1}-\emptyset_{2}\right) & \cdots & sin\left(\emptyset_{1}-\emptyset_{n}\right)\\ sin\left(\emptyset_{2}-\emptyset_{1}\right) & sin\left(\emptyset_{2}-\emptyset_{2}\right) & \cdots & sin\left(\emptyset_{2}-\emptyset_{n}\right)\\ \cdots & \cdots & \cdots & \cdots \\ sin\left(\emptyset_{n}-\emptyset_{1}\right) & sin\left(\emptyset_{n}-\emptyset_{2}\right) & \cdots & sin\left(\emptyset_{n}-\emptyset_{n}\right) \end{array}\right]$$

Through the above steps, one-dimensional time series signals can be transformed into two-dimensional feature images while preserving the temporal correlations between data points. This method provides different levels of information granularity for deep learning models, thereby enhancing the accuracy of classification tasks to a certain extent. The GAF transformation process is illustrated in Figure 1.

### 2.2. ECA Mechanism

To achieve a lightweight model, the Efficient Channel Attention (ECA) mechanism [44] is utilized to enhance the overall performance of the model. ECA is an efficient channel attention mechanism that not only reduces the model’s parameter count, avoiding complex dimensionality transformations, but also enhances the model’s feature representation capability by adaptively learning dependencies between different channels. The diagram of the ECA structure is shown in Figure 2:

From the structure diagram of ECA, it can be observed that its advantage lies in achieving local cross-channel information interaction. Firstly, the aggregate features of each channel are obtained by global average pooling (GAP). Secondly, information exchange across channels is realized by 1D convolution. Finally, the Sigmoid function (σ in the figure) is used to calculate the weight of each channel and weight it with the input features. The size of the convolution kernel determines the scope of the interaction, and there is a nonlinear mapping between it and each channel to ensure that the size of the convolution kernel *K* is adaptive. Inter-channel feature interaction is achieved through Equation (6) [45]. Equation (7) [46] defines the size of the convolutional kernel *K*, where γ and *b* are hyperparameters, set to 2 and 1, respectively.
(7)$$K=\Psi \left(C\right)$$
(8)$$K={\left|\frac{{log}_{2}\left(C\right)}{\gamma }+\frac{b}{\gamma }\right|}_{odd}$$

ECA avoids complex global pooling operations by enabling local information interaction and adaptive weight adjustment, thereby achieving more efficient channel attention computation. Therefore, embedding ECA into residual networks enhances the model’s focus on critical features, making the residual network more effective in extracting fault features.

### 2.3. Improved Lightweight ResNet Model

The ResNet model [47] is developed based on CNN to address the problems of gradient explosion and vanishing gradients as the network depth increases. Its advantage lies in using shortcut connections to stack multiple residual blocks, thereby ensuring the extraction of features from deeper layers. To achieve a lightweight improvement, this manuscript selects the ResNet-18 model for the lightweight design. The residual blocks in ResNet-18 consist of two types: standard residual blocks and residual blocks with downsampling layers. The specific structure is shown in Figure 3.

Due to the relatively large number of parameters in the ResNet-18 model, it may encounter challenges in specific tasks, potentially leading the model to get stuck in local optima and thus not fully realize its optimal performance. To address the aforementioned issues, this manuscript introduces refined modifications to the standard residual block by removing the second convolutional layer. The goal is to simplify the model structure and reduce computational complexity without significantly impacting the feature extraction capability. Additionally, an Efficient Channel Attention (ECA) module is added after each residual block to enhance the model’s sensitivity to important features and its ability to allocate weights effectively. Traditional convolutional layers have been replaced with depthwise separable convolutions (DSConv). Compared to conventional convolutions, DSConv significantly reduces the number of parameters and computational costs, lowering the overall computational burden of the model. Furthermore, DSConv is capable of more deeply capturing critical information from signals, ensuring the robustness and accuracy of the model when handling complex data. After these improvements, the final lightweight model is named E-ResNet13. The modified residual block structure is illustrated in Figure 4, and the overall architecture of E-ResNet13 is depicted in Figure 5. The specific parameters of the model are shown in Table 1.

First, the feature map with dimensions 3 × 224 × 224 is input into three convolutional layers, reducing the feature map size to 64 × 112 × 112. Next, it is passed through a max pooling layer for dimensionality reduction, resulting in an output dimension of 64 × 56 × 56. Subsequently, the feature map undergoes further processing through four improved residual modules. The first residual block consists entirely of standard residual blocks, while the second, third, and fourth residual blocks are stacked with downsampling residual blocks and standard residual blocks. As detailed in Table 1, when the feature map reaches stage 2, the first residual block reduces the feature map size to half of its original dimension and incorporates a downsampling residual structure to ensure equivalent addition of feature map dimensions. Stages 3 and 4 follow this pattern. Finally, the feature map undergoes average pooling (Avgpool) and is classified into fault types using a fully connected (FC) layer.

## 3. Fault Diagnosis Method of Rolling Bearing Based on GAF-E-ResNet13

In this paper, a GAF-E-ResNet13 method is proposed based on the advantages of GAF images, such as the preservation of signal time correlation and the powerful capability of the ResNet model in processing image tasks. By implementing a lightweight design for the ResNet-18 model, not only are the training time costs reduced, but the iteration convergence speed is also increased, thereby improving the model’s recognition accuracy. The overall fault diagnosis process is shown in Figure 6.

Step1: Firstly, a 50% overlapping sampling method is used, extracting 512 data points each time to generate GAF images of size 224 × 224. For each type of fault, 600 GAF images are generated to ensure the diversity and sufficiency of the samples. Secondly, the generated image data are divided into training and testing sets, providing the model with high-quality data input. Finally, these are input into the lightweight model for training and testing, and the completed results and data are saved for subsequent analysis.

Step2: The lightweight E-ResNet13 model is constructed, incorporating the ECA mechanism and improved residual blocks to ensure its ability to retain deep feature extraction while reducing the number of parameters and computational cost. The model finally outputs the fault categories through a fully connected layer, completing the fault identification process.

Step3: After training is completed, the accuracy and loss curves are plotted over iterations, as well as the confusion matrix and a 3D visualization of feature extraction. By analyzing these results, the effectiveness of this method in fault diagnosis is validated, addressing the issues of traditional models with high parameter counts and computational complexity. This approach holds significant research value and promising application prospects.

## 4. Experimental Analysis of Rolling Bearing Fault Diagnosis

The entire experiment was conducted on a Windows 10 system using PyCharm (Version: 2020.1) software and the PyTorch(Version: 1.13.0+cu117) deep learning framework. Training and testing were performed on an RTX 4070 GPU device (NVIDIA, Santa Clara, CA, USA). During the experiment, the Adam optimizer was used to update training parameters adaptively, and the cross-entropy loss function was utilized for calculation.

### 4.1. CWRU Dataset

To validate the effectiveness of the proposed method, this manuscript uses the rolling bearing fault dataset from Case Western Reserve University for verification. The experimental setup is depicted in Figure 7. The rolling bearings used in the experiment are SKF6205-2RS deep groove ball bearings. The bearing faults were introduced using electrical discharge machining, and the fault types include inner race fault, outer race fault, and rolling element fault, with fault diameters of 0.18 mm, 0.36 mm, and 0.53 mm, respectively, totaling nine types of faults. In the experiment, signals with a sampling frequency of 12 kHz were selected for analysis. Tests were conducted under four different operating conditions, ranging from 0 to 3 hp (1 hp = 735 W).

The one-dimensional signals of ten different fault conditions (including normal and nine different faulty bearings) were transformed into GAF images, and the data were labeled accordingly. For each operating condition, each fault type comprises 600 samples, totaling 6000 samples. These samples were split into training and test sets in a 2:1 ratio. The detailed division of the bearing fault dataset and the label settings are shown in Table 2.

### 4.2. Generating GAF Images

In the experiments conducted in this article, each sample consists of 512 data points, and data augmentation was performed using a 50% resampling method to further ensure temporal correlation between signals. Figure 8 displays the time-domain signal plots for different fault conditions, where 1, 2, and 3 represent damage extents of 0.18 mm, 0.36 mm, and 0.54 mm, respectively. The generated GASF and GADF images for different faults are shown in Figure 9 and Figure 10.

### 4.3. Evaluation Metrics

To evaluate the performance of the model in fault classification, this manuscript utilizes four evaluation metrics: accuracy (Acc), precision (Pre), recall (Rec), and F1-score [48]. Accuracy represents the proportion of correctly classified samples out of the total number of samples, and the formula for accuracy is given in Equation (9).
(9)$$\mathrm{Accuracy}=\frac{\mathrm{TP}+\mathrm{TN}}{\mathrm{TP}+\mathrm{TN}+\mathrm{FP}+\mathrm{FN}}$$
where TP (true positive) represents the number of correctly classified positive samples, TN (true negative) denotes the number of correctly classified negative samples, FP (false positive) refers to negative samples incorrectly classified as positive, and FN (false negative) indicates positive samples incorrectly classified as negative.

The formulas for precision and recall are given in Equations (10) and (11), respectively. Precision represents the proportion of predicted positive samples among the total predicted positives, while recall indicates the proportion of correctly predicted positive samples out of the total actual positives. These two metrics focus on evaluating the classification performance for positive samples.
(10)$$\mathrm{Precision}=\frac{\mathrm{TP}}{\mathrm{TP}+\mathrm{FP}}$$
(11)$$\mathrm{Recall}=\frac{\mathrm{TP}}{\mathrm{TP}+\mathrm{FN}}$$

The F1-score is a balanced metric that represents the harmonic mean of precision and recall, providing a comprehensive analysis by balancing the trade-off between the two. The F1-score reflects the equilibrium between precision and recall, which is crucial for redefining their combined performance. The formula for the F1-score is given in Equation (12).
(12)$$F_{1-\mathrm{score}}=\frac{2\times \mathrm{Precision}\times \mathrm{Recall}}{\mathrm{Precision}+\mathrm{Recall}}$$

### 4.4. Comparative Analysis

#### 4.4.1. Comparative Experiment Analysis between GADF and GASF

To compare the advantages of the two GAF image encoding methods, dataset A was used for verification. GADF and GASF were respectively used as inputs to the lightweight model. The batch size for the training set was set to 32, while the batch size for the test set was set to 16, with a learning rate of 0.0001. After training for 120 epochs, the accuracy and loss curves are shown below.

According to the curves in Figure 11, both GADF and GASF methods perform excellently on the model proposed in the article, with accuracy reaching convergence by the 20th epoch for both methods. Notably, the training process for GADF is more stable compared to GASF, exhibiting smaller fluctuations, with an average accuracy of 98.75% and a loss rate approaching zero. This is because GADF images tend to highlight more features and reflect the differences between various states more effectively, enabling the model to more easily capture the information in the images. The accuracy, precision, recall, and F1-score calculated using Equations (9)–(12) are shown in Table 3.

As shown in Table 3, the evaluation metrics for GADF are superior to those for GASF, indicating that GADF images have a stronger feature representation capability when used as model inputs. However, this manuscript chooses to use the relatively lower-performing GASF method for further comparative analysis. This approach aims to enhance the persuasiveness and interpretability of the research by providing a thorough analysis of the GASF method. It reveals the potential advantages and limitations of GASF in specific contexts, while also validating the superiority of GADF.

#### 4.4.2. Comparison of Different Image Encoding Methods

To validate the superiority of GAF image encoding, several commonly used image encoding methods, including Markov Transition Field (MTF), Continuous Wavelet Transform (CWT), Recurrence Plot (RP), and traditional grey-scale images, are employed as model inputs and compared with GASF. These methods use dataset A for generation, with sliding resampling applied to produce images of size 224 × 224. To maintain data integrity while capturing the key features of the signals, the step sizes for MTF and RP are set to 256, the step size for CWT is set to 180, and the sampling step size for grey-scale images is set to 50. In the experiments, the dataset is divided according to Table 2 and trained for 50 iterations. The resulting test accuracy curves are shown in Figure 12.

As shown in the figure, the accuracy for MTF, RP, and CWT methods all exhibit an increasing trend, but their convergence speed and maximum accuracy values are lower than those of the GASF method. Additionally, the grey-scale map shows excessive fluctuation, resulting in non-convergence of the curve. This is because the grey-scale encoding method fails to effectively represent the fault information contained in one-dimensional signals. Overall, GASF is demonstrated to effectively represent information about different rolling bearing faults, validating its feasibility. The bar chart displaying the four evaluation metrics for the different encoding methods is shown in Figure 13.

The bar chart clearly shows that the GASF encoding method outperforms the other four methods across all four evaluation metrics, demonstrating its superiority in fault diagnosis. Notably, the evaluation metrics for both the grey-scale map and CWT encoding methods are generally below 80%, indicating deficiencies in feature extraction and an inability to fully represent the fault information in one-dimensional signals. Overall, the GASF method exhibits superior performance in rolling bearing fault identification.

#### 4.4.3. Comparison of Different Intelligent Algorithm Methods

To validate the superiority of the proposed method, we conducted comparative experiments using several popular deep learning models, including ConvNeXt-T, ResNet-18, DenseNet-121, the unmodified lightweight model ResNet-13, and the recently popular GhostNet lightweight model. All models were trained for 50 epochs with GASF as the input. To thoroughly assess the performance and efficiency of the proposed method, we calculated not only the training and testing times for each model but also the number of model parameters and computational operations. The iterative parameters for each model are shown in Table 4.

As shown in the table, compared with other models, the proposed lightweight model demonstrates shorter training and testing times. Among the models, GhostNet has the lowest computational complexity at only 155.86 M, though its parameter count is lower than that of our model, which has 1.15 M parameters. To further evaluate the model’s performance, we calculated the evaluation metrics for the test set, with the results depicted in Figure 14. The ConvNeXt-T model had the lowest metric values, likely due to its overly deep architecture, which led the feature extraction process to become trapped in local optima. In contrast, the GhostNet model outperformed the other four models in most aspects; however, its longer run time and higher parameter count resulted in slower convergence. Overall, the comprehensive analysis shows that the proposed lightweight model performs the best, validating the feasibility and efficiency of the lightweight design.

Figure 15 displays the classification results of different models in the confusion matrix. The x-axis represents the predicted values, and the y-axis represents the true values. The main diagonal indicates the number of correctly classified samples. Categories 0 through 9 correspond to the fault levels of rolling elements (three fault levels), inner race (three fault levels), outer race (three fault levels), and normal bearings. The other models, having deeper layers and higher complexity, show slight overfitting during testing, which affects their accuracy. As illustrated, the proposed method correctly classifies the largest number of samples, demonstrating superior feature learning capability compared to the other models.

### 4.5. Transfer Experiments and Analysis under Different Operating Conditions

To further validate the generalization capability of the proposed model, migration experiments were conducted using datasets B, C, and D under varying operating conditions. Each dataset was used as the source domain for training and the other two datasets as target domains for testing, with 50 iterations conducted for each condition. For example, if dataset B is set as the source domain for training and dataset C as the target domain for testing, this is denoted as “B → C”. The migration test results are shown in Table 5, with all transfer conditions achieving metrics above 97%. These results confirm the generalization and robustness of the proposed model.

To observe the feature extraction effectiveness more clearly, T-SNE clustering visualization was used to create dimensionality reduction plots for the six migration scenarios. Figure 16 visually displays the classification performance of the proposed model in three-dimensional space under varying operating conditions. The results indicate that the proposed model, combined with GASF, exhibits strong feature learning capabilities. Different colors in the figure represent different faults, and it can be seen that most of the features for the ten types of rolling bearing faults are correctly distinguished, validating the feature extraction capability of the proposed model under different conditions.

To verify that the proposed model has stronger generalization capability under varying conditions compared to other algorithms, two methods with performance metrics close to those of the proposed method were selected for comparison: GhostNet and DenseNet-121. These methods were each tested across the six migration scenarios with 50 iterations. The results are shown in Table 6 and Table 7.

The data from the tables indicate that the four evaluation metrics for the six different migration scenarios of the proposed method are all above 97%, and outperform the evaluation metrics of GhostNet and DenseNet-121 under various conditions. These migration results confirm that the proposed method exhibits good generalization performance across different operating conditions. To provide a more intuitive reflection of the feature extraction effectiveness of the other two models under different conditions, T-SNE three-dimensional visualization plots were created. These T-SNE plots further illustrate the superiority of the proposed model in migration across different conditions. The specific T-SNE plots are shown in Figure 17 and Figure 18:

## 5. Conclusions

To address issues such as the large number of parameters, extended runtime, and high memory usage associated with the ResNet-18 model, the article proposes a lightweight fault diagnosis model named E-ResNet13. This model incorporates improvements to the residual blocks and integrates the ECA mechanism. Combining the GAF image encoding method with the proposed model, experiments were conducted on the CWRU bearing fault dataset, validating the effectiveness of the proposed method. The main conclusions are as follows:(1)Compared with the other four encoding methods—MTF, CWT, RP, and grey-scale map—the proposed GAF image encoding method demonstrates a superior ability to represent differences in internal fault characteristics of vibration signals under various fault modes. Specifically, the GASF image encoding method achieves an accuracy of 95.73%, precision of 95.88%, recall of 95.73%, and F1-score of 95.76%, all exceeding the performance of the other image encoding methods by more than 10%. Thus, it can be seen that the use of two-dimensional images overcomes the limitations of feature extraction from one-dimensional signals.(2)Experimental validation shows that the proposed lightweight E-ResNet13 model exhibits high superiority. Compared with other intelligent diagnostic algorithms, the proposed model has the fewest parameters and computational requirements, with shorter training and testing times. This confirms the effectiveness of the model’s lightweight design. Specifically, the training time for the model is 630.95 s, the testing time is 242.39 s, the number of computations is 1110.22 M, and the number of parameters is 1.15 M. The model designed in this paper achieves high fault recognition accuracy while ensuring minimal computational and parameter requirements.(3)In the migration tests under different operating conditions, the proposed E-ResNet13 model demonstrates high generalization capability. Specifically, the average values of the evaluation metrics for the six migration scenarios of the E-ResNet13 model are 98.23% accuracy, 98.26% precision, 98.23% recall, and an F1-score of 98.23%. Compared to the GhostNet model, the E-ResNet13 model surpasses it in the four evaluation metrics by 1.59%, 1.62%, 1.59%, and 1.60%, respectively. Compared to the DenseNet-121 model, the E-ResNet13 model exceeds its performance in the metrics by 4.23%, 4.04%, 4.23%, and 4.18%, respectively.

## Figures and Tables

**Figure 1 sensors-24-05952-f001:**
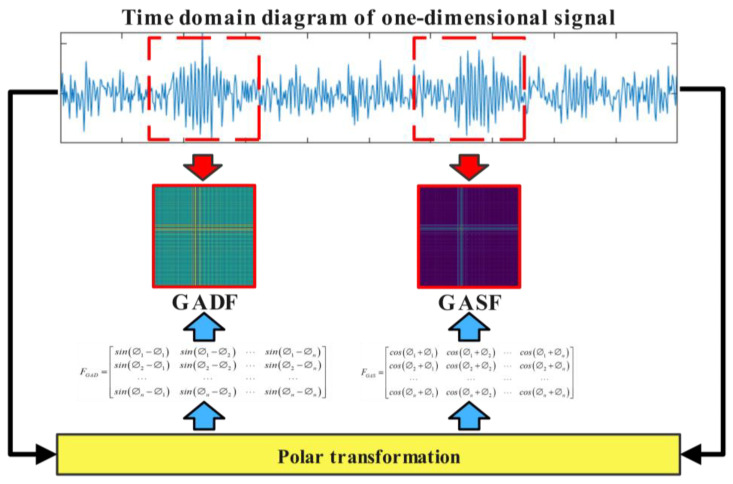
Flowchart of GAF encoding.

**Figure 2 sensors-24-05952-f002:**
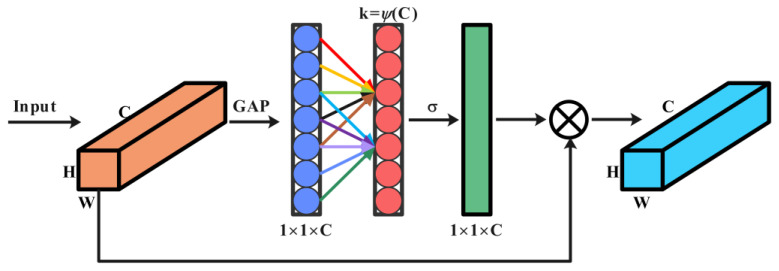
ECA structure diagram.

**Figure 3 sensors-24-05952-f003:**
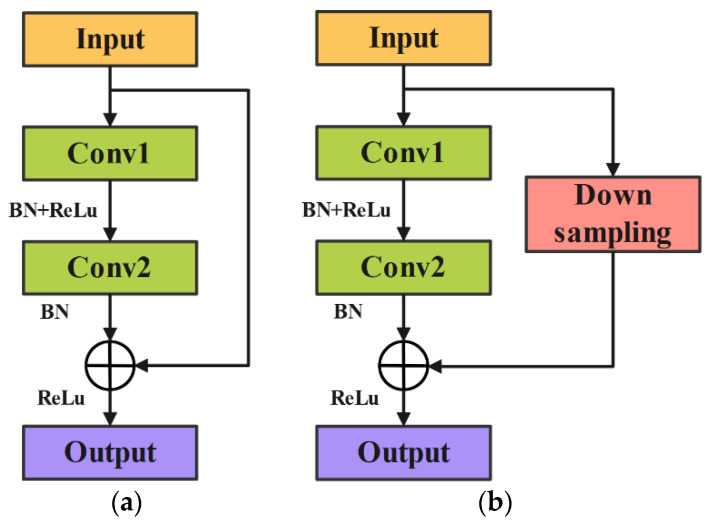
Two different residual blocks. (**a**) Standard residual blocks; (**b**) residual blocks with downsampling layers.

**Figure 4 sensors-24-05952-f004:**
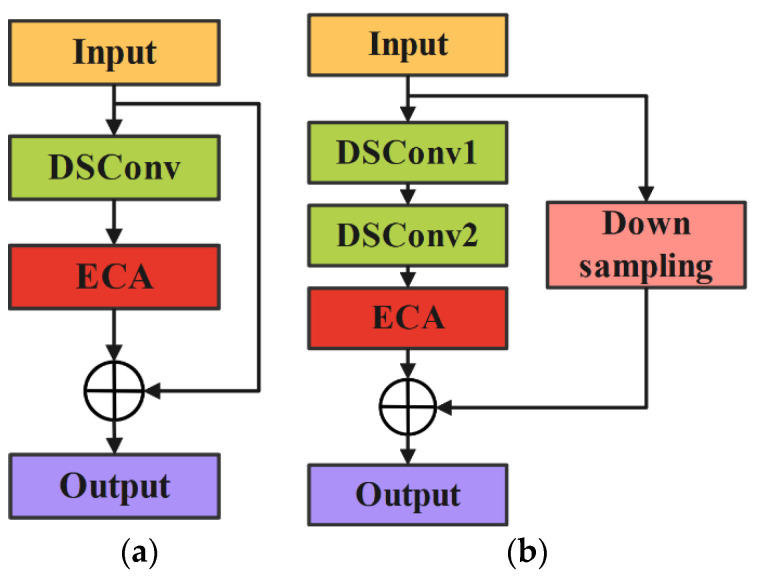
Improved residual block. (**a**) Improved module 1; (**b**) improved module 2.

**Figure 5 sensors-24-05952-f005:**
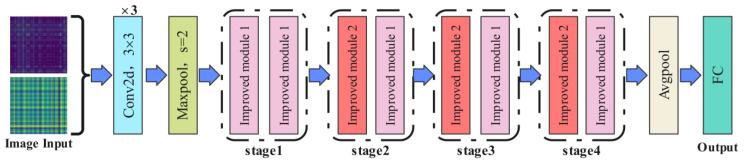
Improved model structure diagram.

**Figure 6 sensors-24-05952-f006:**
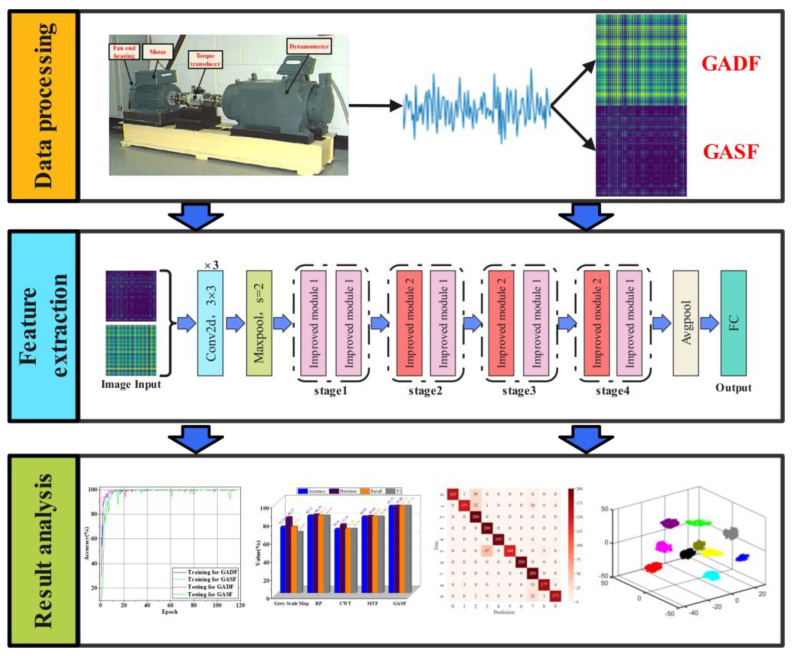
The rolling bearing fault diagnosis flowchart.

**Figure 7 sensors-24-05952-f007:**
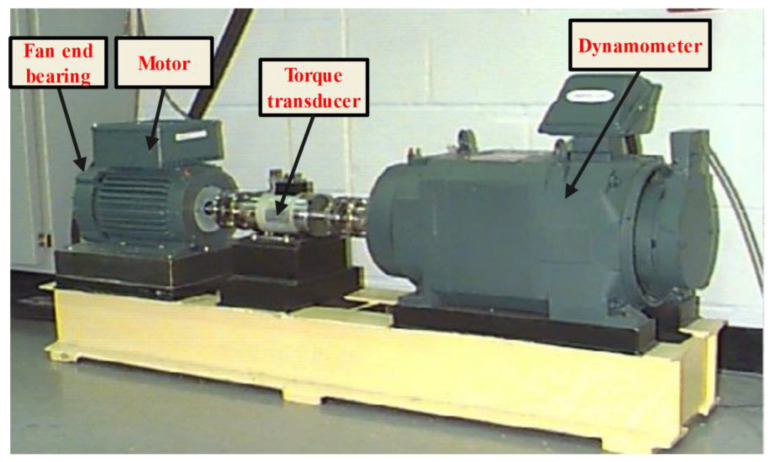
CWRU test bench.

**Figure 8 sensors-24-05952-f008:**
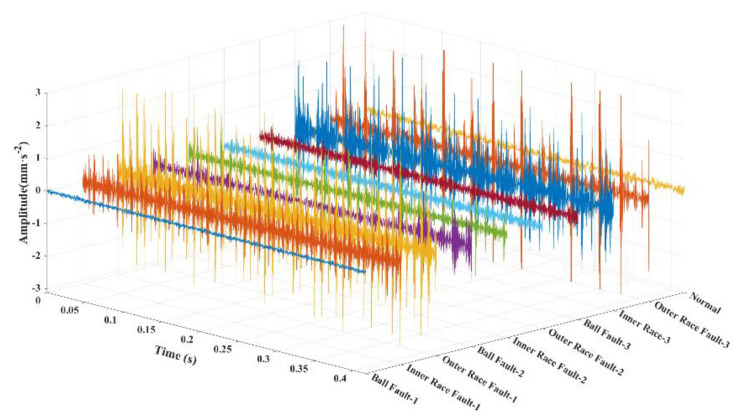
Time domain diagram of signals of different bearing faults.

**Figure 9 sensors-24-05952-f009:**
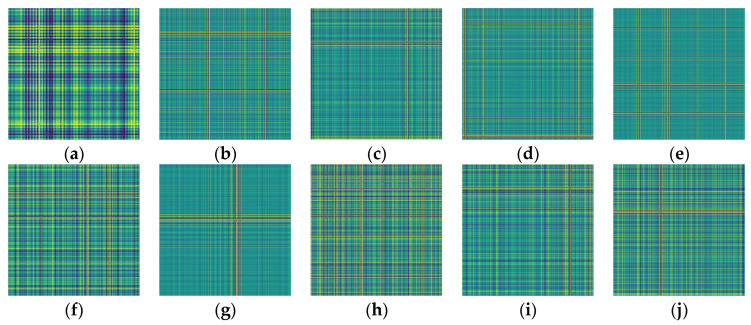
GASF images of 10 bearing fault conditions. (**a**) Normal bearing; (**b**) Inner race fault-1; (**c**) Inner race fault-2; (**d**) Inner race fault-3; (**e**) Outer race fault-1; (**f**) Outer race fault-2; (**g**) Outer race fault-3; (**h**) Ball fault-1; (**i**) Ball fault-2; (**j**) Ball fault-3.

**Figure 10 sensors-24-05952-f010:**
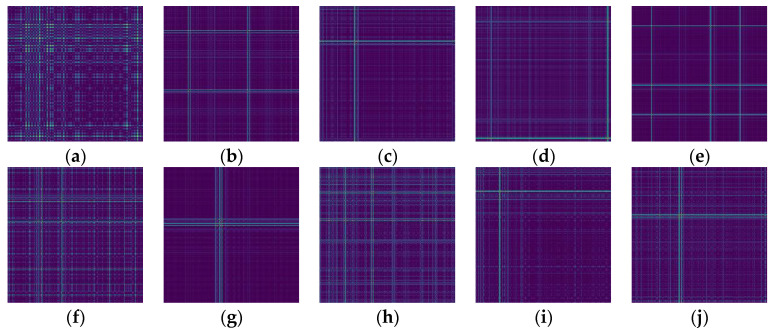
GADF images of 10 bearing fault conditions. (**a**) Normal bearing; (**b**) Inner race fault-1; (**c**) Inner race fault-2; (**d**) Inner race fault-3; (**e**) Outer race fault-1; (**f**) Outer race fault-2; (**g**) Outer race fault-3; (**h**) Ball fault-1; (**i**) Ball fault-2; (**j**) Ball fault-3.

**Figure 11 sensors-24-05952-f011:**
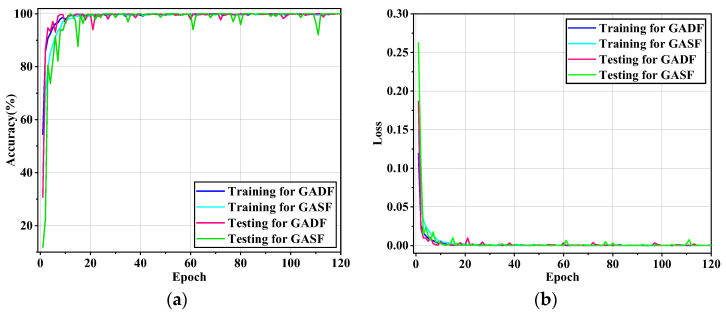
(**a**) Accuracy curve and (**b**) loss curve.

**Figure 12 sensors-24-05952-f012:**
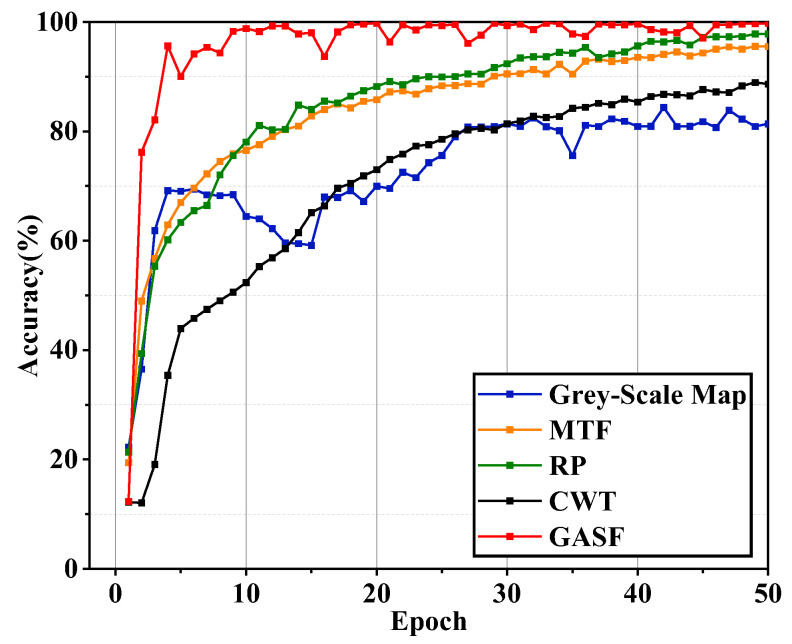
Iteration curves of different coding methods.

**Figure 13 sensors-24-05952-f013:**
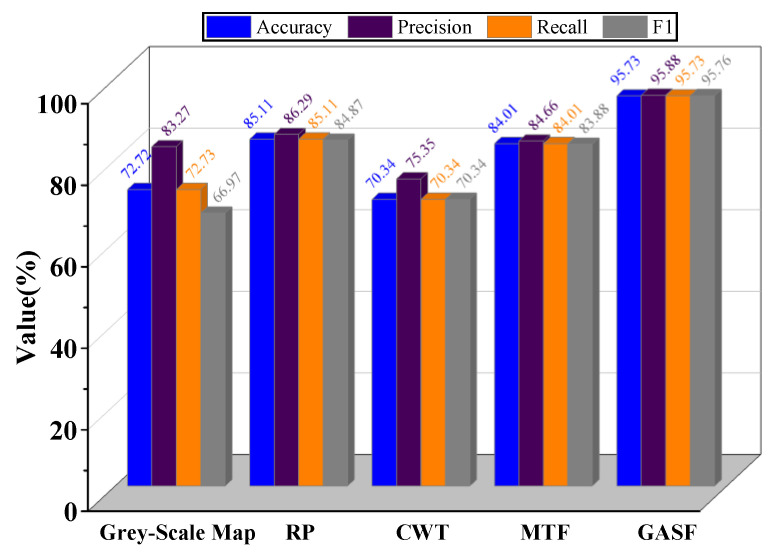
Evaluation metrics for different encoding methods.

**Figure 14 sensors-24-05952-f014:**
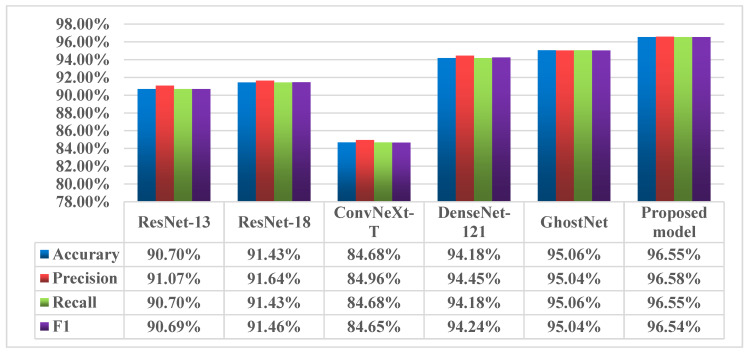
Evaluation metrics for different models.

**Figure 15 sensors-24-05952-f015:**
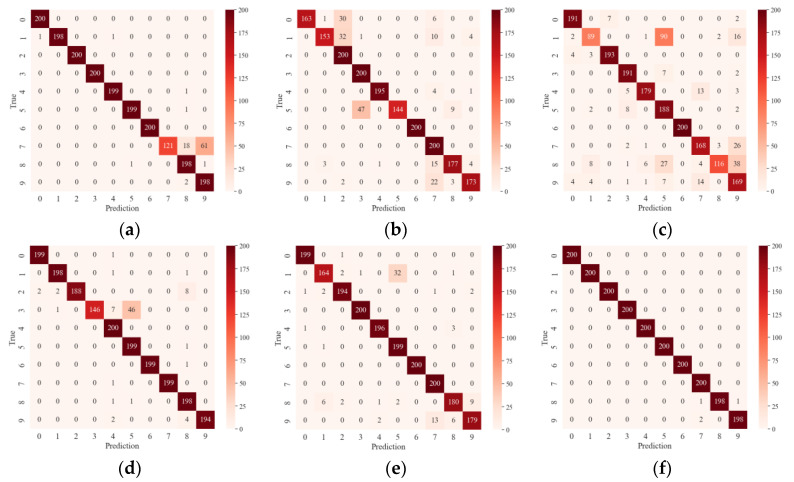
Confusion matrix (**a**) ResNet-13; (**b**) ResNet-18; (**c**) ConvNeXt-T; (**d**) DenseNet-121; (**e**) GhostNet; (**f**) proposed model.

**Figure 16 sensors-24-05952-f016:**
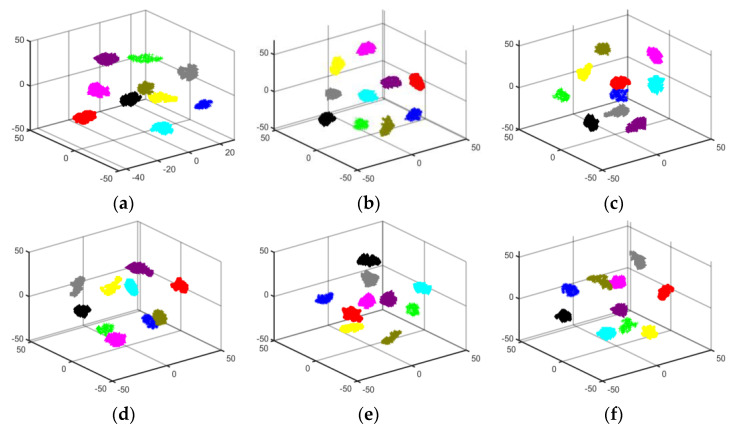
T-SNE plots for different migration scenarios of the proposed method: (**a**) B → C; (**b**) B → D; (**c**) C → B; (**d**) C → D; (**e**) D → B; (**f**) D → C.

**Figure 17 sensors-24-05952-f017:**
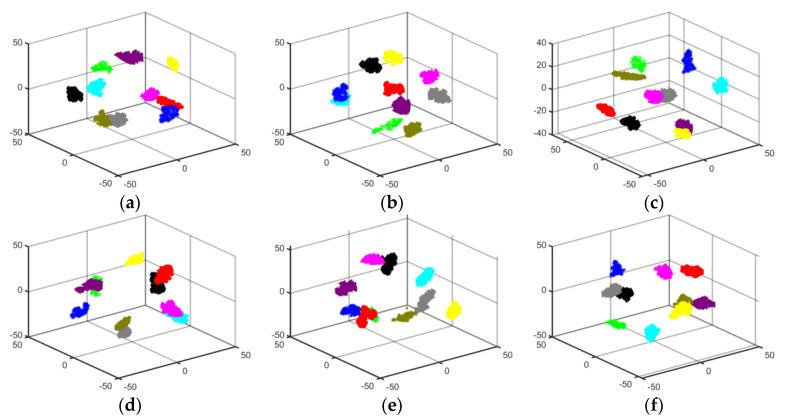
T-SNE plots for different migration scenarios of DenseNet-121: (**a**) B → C; (**b**) B → D; (**c**) C → B; (**d**) C → D; (**e**) D → B; (**f**) D → C.

**Figure 18 sensors-24-05952-f018:**
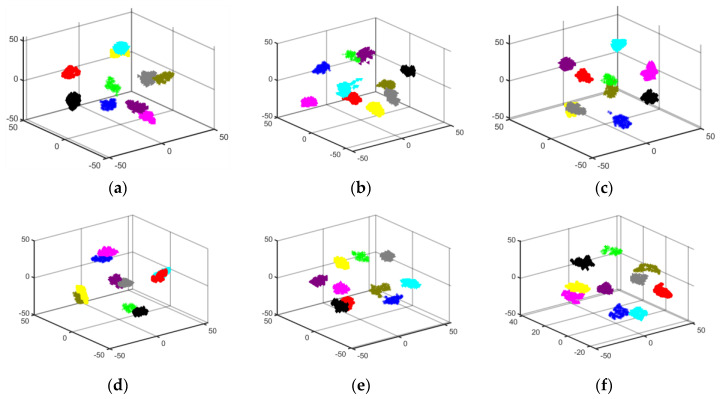
T-SNE plots for different migration scenarios of GhostNet: (**a**) B → C; (**b**) B → D; (**c**) C → B; (**d**) C → D; (**e**) D → B; (**f**) D → C.

**Table 1 sensors-24-05952-t001:** Parameter settings for the improved model.

Number	Layout	Parameter Setting	Output Dimension
1	Input	-	3 × 224 × 224
2	3*Conv2d	$\left[\begin{array}{l} K=3,S=2,64\\ K=3,S=1,64\\ K=3,S=1,64 \end{array}\right]$	64 × 112 × 112
3	Maxpool	*K* = 3, *S* = 2, 64	64 × 56 × 56
4	stage1	$\left[\begin{array}{c} DSConv:K=3,S=1,64\\ \begin{array}{c} ECA\\ \begin{array}{c} DSConv:K=3,S=1,64\\ ECA \end{array} \end{array} \end{array}\right]$	64 × 56 × 56
5	stage2	$\left[\begin{array}{c} \begin{array}{c} DSConv1:K=3,S=2,128\\ DSConv2:K=3,S=1,128 \end{array}\\ ECA\\ \begin{array}{c} DSConv:K=3,S=1,128\\ ECA \end{array} \end{array}\right]$	128 × 28 × 28
6	stage3	$\left[\begin{array}{c} \begin{array}{c} DSConv1:K=3,S=2,256\\ DSConv2:K=3,S=1,256 \end{array}\\ ECA\\ \begin{array}{c} DSConv:K=3,S=1,256\\ ECA \end{array} \end{array}\right]$	256 × 14 × 14
7	stage4	$\left[\begin{array}{c} \begin{array}{c} DSConv1:K=3,S=2,512\\ DSConv2:K=3,S=1,512 \end{array}\\ ECA\\ \begin{array}{c} DSConv:K=3,S=1,512\\ ECA \end{array} \end{array}\right]$	512 × 7 × 7
8	Avgpool	-	512 × 1 × 1
9	FC	-	512, 1000

**Table 2 sensors-24-05952-t002:** Dataset division and label setting of rolling bearings.

Damage Degree/mm	0.18			0.36			0.54			0	Load
	Inner Race	Outer Race	Ball	Inner Race	Outer Race	Ball	Inner Race	Outer Race	Ball	Normal	
Labels	0	1	2	3	4	5	6	7	8	9	
A	Training	400	400	400	400	400	400	400	400	400	0
	Test	200	200	200	200	200	200	200	200	200	
B	Training	400	400	400	400	400	400	400	400	400	1 hp
	Test	200	200	200	200	200	200	200	200	200	
C	Training	400	400	400	400	400	400	400	400	400	2 hp
	Test	200	200	200	200	200	200	200	200	200	
D	Training	400	400	400	400	400	400	400	400	400	3 hp
	Test	200	200	200	200	200	200	200	200	200	

**Table 3 sensors-24-05952-t003:** Evaluation metrics for GASF and GADF.

Method	Accuracy	Precision	Recall	F1
GASF	97.18%	97.26%	97.18%	97.20%
GADF	98.75%	98.75%	98.74%	98.75%

**Table 4 sensors-24-05952-t004:** Comparison of parameters for different models.

Model	Training Time (s)	Test Time (s)	Flops (M)	Params (M)
ResNet-13	1065.22	316.21	4972.75	8.01
ResNet-18	640.34	249.44	1824.03	11.69
ConvNeXt-T	1323.79	326.72	4454.77	27.81
DenseNet-121	1570.02	334.25	2895.79	6.96
GhostNet	1207.47	277.70	155.86	5.18
Proposed model	630.95	242.39	1110.22	1.15

**Table 5 sensors-24-05952-t005:** Migration results for different operating conditions.

Different Migration Schemes	Accuracy (%)	Precision (%)	Recall (%)	F1 (%)
B → C	98.85	98.85	98.85	98.84
B → D	98.11	98.25	98.12	98.14
C → B	97.59	97.60	97.59	97.59
C → D	98.75	98.75	98.75	98.75
D → B	97.91	97.91	97.91	97.89
D → C	98.16	98.17	98.16	98.15

**Table 6 sensors-24-05952-t006:** Migration scenarios for GhostNet under different operating conditions.

GhostNet Different Migration Schemes	Accuracy (%)	Precision (%)	Recall (%)	F1 (%)
B → C	95.68	95.67	95.68	95.67
B → D	95.73	95.76	95.73	95.75
C → B	96.94	96.94	96.94	96.94
C → D	98.25	98.26	98.25	98.25
D → B	95.22	95.25	95.22	95.21
D → C	97.99	97.98	97.99	97.98

**Table 7 sensors-24-05952-t007:** Migration scenarios for DenseNet-121 under different operating conditions.

DenseNet-121 Different Migration Schemes	Accuracy (%)	Precision (%)	Recall (%)	F1 (%)
B → C	96.92	96.95	96.92	96.92
B → D	95.20	95.36	95.20	95.28
C → B	93.42	93.49	93.42	93.42
C → D	95.36	95.50	95.36	95.38
D → B	89.19	89.90	89.19	89.38
D → C	93.89	94.10	93.89	93.94

## Data Availability

The original data presented in the study are openly available in Case Western Reserve University Bearing Data. https://engineering.case.edu/bearingdatacenter.

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
