# Peer review of "Research on Fault Diagnosis of Rolling Bearing Based on Gramian Angular Field and Lightweight Model"

_sensors, 2024, doi:10.3390/s24185952_

Round 1

Reviewer 1 Report

Comments and Suggestions for Authors

This manuscript presents a fault diagnosis method of rolling bearing based on gramian angular field and lightweight model. Some suggestions are below.

1. The summary of existing research in the introduction section of the manuscript is insufficient and needs further improvement.

2. The writing standards of the manuscript still need to be improved, and the reviewers have identified some obvious errors.

3. Almost all the formulas in the manuscript have no references, so it is not a problem if they are all proposed by the author. However, if the formula is based on the work of others, corresponding literature must be provided.

4. The method proposed in the manuscript contains numerous key parameters that have a significant impact on the performance of the algorithm. However, the author did not provide principles on how to select these parameters.

5. Figure 3(b) adds downsampling to the standard residual network. Downsampling causes the feature maps to become smaller, making them inconsistent with the size of the main feature maps, so how can the smaller feature maps be added directly to the main stem?

6. The two subgraphs in Fig. 4 are clearly different, while both (a) and (b) are described as "Improving module 1", which will obviously confuse the reader.

7. In Table 3, why the training and testing time of ResNet18 is shorter than the proposed model, while the number of parameters and FLOPs are larger?

8. Moreover, the literature review of this manuscript is not comprehensive. In the introduction, the author's summary of existing fault diagnosis methods is not enough. I hope that the authors can add some new references in order to improve the reviews and the connection with the literatures:

[1] Transformer and Graph Convolution-Based Unsupervised Detection of Machine Anomalous Sound Under Domain Shifts[J]. IEEE Transactions on Emerging Topics in Computational Intelligence, 2024:1-16.

Comments on the Quality of English Language

2. The writing standards of the manuscript still need to be improved, and the reviewers have identified some obvious errors.

Author Response

Dear reviewer:

I’m very grateful to your comments for the manuscript (Number: sensors-3139599, Title: Research on fault diagnosis of rolling bearing based on gramian angular field and lightweight model. Your comments are very important and helpful to my thesis writing and study. We have studied the valuable comments from you carefully, and tried our best to revise the manuscript. The manuscript has undergone English language editing by MDPI. The point to point responds to your comments are listed as following:

Point 1: The summary of existing research in the introduction section of the manuscript is insufficient and needs further improvement.

Response 1: Thank you for your question. We have improved the discussion of research methods in the introduction, and added references [17][18][25][26][32][33] to ensure the adequacy of the article.

Point 2: The writing standards of the manuscript still need to be improved, and the reviewers have identified some obvious errors.

Response 2: Thank you for your question. We have carefully checked the full text for English editing errors and have corrected the errors.

Point 3: Almost all the formulas in the manuscript have no references, so it is not a problem if they are all proposed by the author. However, if the formula is based on the work of others, corresponding literature must be provided.

Response 3: Thank you for your question. We transposed the reference [34][35] before the formula and added the reference [37][38] for the formula in section 2.2.

Point 4: The method proposed in the manuscript contains numerous key parameters that have a significant impact on the performance of the algorithm. However, the author did not provide principles on how to select these parameters.

Response 4: Thank you for your question. Due to my previous negligence, I failed to adequately express the parameters of the model. We re-learned the model parameter changes and constructed Table 1 for readers' convenience.

Point 5: Figure 3(b) adds downsampling to the standard residual network. Downsampling causes the feature maps to become smaller, making them inconsistent with the size of the main feature maps, so how can the smaller feature maps be added directly to the main stem?

Response 5: Thank you for your question.In the end of Table 1, we clearly explain the dimensional changes of the feature graph in the model.

Point 6: The two subgraphs in Fig. 4 are clearly different, while both (a) and (b) are described as "Improving module 1", which will obviously confuse the reader.

Response 6: Thank you for your question. Sorry, this problem was caused by my carelessness, and I have revised it in the revised draft.

Point 7: In Table 3, why the training and testing time of ResNet18 is shorter than the proposed model, while the number of parameters and FLOPs are larger?

Response 7: Thank you for your question. Our aim is to reduce the number of parameters in the model and the amount of computation involved. After the model is built, there may be some internal lightweight structural redundancy, resulting in poor program and hardware performance, resulting in high training and test times. I have optimized the structure of the model. Because the results of each operation are different, we conducted five experiments and took the average value of the running time, the specific values are shown in Table 4. In the following paper, we also compare the classification accuracy and find that the proposed method is higher than ResNet18.

Point 8: Moreover, the literature review of this manuscript is not comprehensive. In the introduction, the author's summary of existing fault diagnosis methods is not enough. I hope that the authors can add some new references in order to improve the reviews and the connection with the literatures:

[1] Transformer and Graph Convolution-Based Unsupervised Detection of Machine Anomalous Sound Under Domain Shifts[J]. IEEE Transactions on Emerging Topics in Computational Intelligence, 2024:1-16.

Response 8: Thanks for your question, we have added this reference to this article with the specific label [13].

We tried our best to improve the manuscript and made some changes in the manuscript. We appreciate for reviewers’ warm work earnestly, and hope that the correction will meet with approval. Once again, thank you very much for your comments and suggestions. I hope I can have the opportunity to learn and communicate with you in the future.

Reviewer 2 Report

Comments and Suggestions for Authors

This paper proposes a rolling bearing fault diagnosis method based on the Gramian Angular Field (GAF) and an enhanced lightweight residual network, which has certain innovation and practical value.

1、In the abstract, the article points out that "deep learning models have limitations in processing one-dimensional signal feature extraction, and the complexity of the model leads to low training accuracy and high consumption of computing resources." However, the whole article does not provide any theoretical and experimental explanations to support this conclusion. Please supplement it;

2、In the introduction, the article states that "in order to solve the problem of a large number of parameters in the deep learning model", but the whole article does not give the number of parameters of the proposed model or compare it with the existing models. Please supplement the explanation;

3、In section 2.3, the article points out that "this manuscript prioritizes the selection of the ResNet-18 model with the fewest parameters for lightweight design." and "due to the relatively large number of parameters in the ResNet-18 model, it may encounter challenges in specific tasks." Please pay attention to the expression throughout the article to facilitate the reader's understanding and revise the whole text;

4、In section 2.3, the article states that "this manuscript removes the second convolutional layer in the standard residual block and adds an ECA module after the convolutional layers within the residual blocks. In addition, the traditional convolutional layers are replaced by depth-wise separable convolutions (DSConv)." This should be the greatest innovation in the whole article, but the description is extremely inadequate. Please supplement the explanation;

5、Why is there no validation set set up in the experimental part of the article?

Comments on the Quality of English Language

The English of the text can be polished and improved

Author Response

Dear reviewer:

I’m very grateful to your comments for the manuscript (Number: sensors-3139599, Title: Research on fault diagnosis of rolling bearing based on gramian angular field and lightweight model. Your comments are very important and helpful to my thesis writing and study. We have studied the valuable comments from you carefully, and tried our best to revise the manuscript. The manuscript has undergone English language editing by MDPI. The point to point responds to your comments are listed as following:

Point 1: In the abstract, the article points out that "deep learning models have limitations in processing one-dimensional signal feature extraction, and the complexity of the model leads to low training accuracy and high consumption of computing resources." However, the whole article does not provide any theoretical and experimental explanations to support this conclusion. Please supplement it;

Response 1: Thank you for your question. It is explained in conclusions (1) and (2) : 1.Thus, it can be seen that the use of two-dimensional images overcomes the limitations of feature extraction from one-dimensional signals. 2.The model designed in this paper achieves high fault recognition accuracy while ensuring minimal computational and parameter requirements.

Point 2: In the introduction, the article states that "in order to solve the problem of a large number of parameters in the deep learning model", but the whole article does not give the number of parameters of the proposed model or compare it with the existing models. Please supplement the explanation;

Response 2: Thank you for your question. We have analyzed the parameters and calculation amount of the proposed model and other models. The specific data are shown in Table 4 and explained below.

Point 3: In section 2.3, the article points out that "this manuscript prioritizes the selection of the ResNet-18 model with the fewest parameters for lightweight design." and "due to the relatively large number of parameters in the ResNet-18 model, it may encounter challenges in specific tasks." Please pay attention to the expression throughout the article to facilitate the reader's understanding and revise the whole text;

Response 3: Thank you for your question. We have changed ‘’For lightweight improvements, this manuscript prioritizes selects the ResNet-18 model, which has the fewest parameters, for lightweight design’’ to ‘’To achieve a lightweight improvement, this manuscript selects the ResNet-18 model for the lightweight design.’’.

Point 4: In section 2.3, the article states that "this manuscript removes the second convolutional layer in the standard residual block and adds an ECA module after the convolutional layers within the residual blocks. In addition, the traditional convolutional layers are replaced by depth-wise separable convolutions (DSConv)." This should be the greatest innovation in the whole article, but the description is extremely inadequate. Please supplement the explanation;

Response 4: Thank you for your question. We have amended it to: To address the aforementioned issues, this manuscript introduces refined modifications to the standard residual block by removing the second convolutional layer. The goal is to simplify the model structure and reduce computational complexity without significantly impacting the feature extraction capability. Additionally, an Efficient Channel Attention (ECA) module is added after each residual block to enhance the model's sensitivity to important features and its ability to allocate weights effectively. Traditional convolutional layers have been replaced with depthwise separable convolutions (DSConv). Compared to conventional convolutions, DSConv significantly reduces the number of parameters and computational costs, lowering the overall computational burden of the model. Furthermore, DSConv is capable of more deeply capturing critical information from signals, ensuring the robustness and accuracy of the model when handling complex data.

Point 5: Why is there no validation set set up in the experimental part of the article?

Response 5: Thank you for your question. In Section 4.4, we conducted experimental verification and analysis of different coding methods, different intelligent diagnosis algorithms and model migration generalization.

We tried our best to improve the manuscript and made some changes in the manuscript. We appreciate for reviewers’ warm work earnestly, and hope that the correction will meet with approval. Once again, thank you very much for your comments and suggestions. I hope I can have the opportunity to learn and communicate with you in the future.

Reviewer 3 Report

Comments and Suggestions for Authors

I have carefully reviewed the manuscript entitled “Research on fault diagnosis of rolling bearing based on gramian angular field and lightweight model”. This study introduces a novel fault diagnosis method for rolling bearings that leverages the Gramian Angular Field (GAF) and an enhanced lightweight residual network. The research addresses the challenges faced by deep learning models in processing one-dimensional signals, which often result in low training accuracy and high computational resource consumption due to model complexity. Overall, this paper may have some practical innovation, however, there are some limitations, questions and recommendations as follows:

1. The interval [1,1] mentioned in Step 1 should be wrong, please check it.

2. The formula (1) is supposed to be wrong since the range of the normalize value is strange, please check and modify it.

3. The symbol X is different in Step 1 and Step 2 for its expression, please use characters and variables in a standardized manner.

4. The notations following the formula should not contain any spaces; please ensure the format is standardized.

5. In the formula of line 152, the symbols are not explained, and the formula should be separately numbered.

6. The sub-titles of Figure 4 are the same, please check it.

Comments on the Quality of English Language

Minor editing of English language required.

Author Response

Dear reviewer:

I’m very grateful to your comments for the manuscript (Number: sensors-3139599, Title: Research on fault diagnosis of rolling bearing based on gramian angular field and lightweight model. Your comments are very important and helpful to my thesis writing and study. We have studied the valuable comments from you carefully, and tried our best to revise the manuscript. The manuscript has undergone English language editing by MDPI. The point to point responds to your comments are listed as following:

Point 1: The interval [1,1] mentioned in Step 1 should be wrong, please check it.

Response 1: Thank you for your question. We have modified it to [-1,1].

Point 2: The formula (1) is supposed to be wrong since the range of the normalize value is strange, please check and modify it.

Response 2: Thank you for your question. We have modified it. For details, see Formula (1).

Point 3: The symbol is different in Step 1 and Step 2 for its expression, please use characters and variables in a standardized manner.

Response 3: Thank you for your question. We have unified the format of formula symbols.

Point 4: The notations following the formula should not contain any spaces; please ensure the format is standardized.

Response 4: Thank you for your question. We have removed the space after the formula.

Point 5: In the formula of line 152, the symbols are not explained, and the formula should be separately numbered.

Response 5: Thank you for your question. We've edited it separately and explained what it means.

Point 6: The sub-titles of Figure 4 are the same, please check it.

Response 6: Thank you for your question. We have modified it.

We tried our best to improve the manuscript and made some changes in the manuscript. We appreciate for reviewers’ warm work earnestly, and hope that the correction will meet with approval. Once again, thank you very much for your comments and suggestions. I hope I can have the opportunity to learn and communicate with you in the future.

Reviewer 4 Report

Comments and Suggestions for Authors This paper innovatively proposes a rolling bearing fault diagnosis method based on Gramian Angular Field (GAF) and enhanced lightweight residual network. There are many researchers on bearing fault diagnosis method based on neural network, which is also a very meaningful and valuable research field. The research background and theoretical knowledge of the paper are fully introduced, which proves that the author has mastered the latest research progress of bearing fault diagnosis methods at home and abroad. Before the acceptance of the paper, the reviewers believe that there are still some issues that need to be revised or clarified : (1)To address issues such as the large number of parameters, extended runtime, and high memory usage associated with the ResNet-18 model, the article proposes a light weight fault diagnosis model named E-ResNet13. However, the author does not compare the research methods based on the current computing equipment, which is not convincing. (2)For specific industrial application scenarios, do not know what kind of defects will occur in rolling bearings, and there is no pre-fault sample data of rolling bearing inner ring, outer ring, cage, and rolling element. Can your method still be applied ? (3)The improved module 2 is applied in Fig.5, but it is not reflected in Fig.4.On the contrary, two improved modules 1 appear in Fig.4. (4)The paper uses the migration test to verify, but this is only limited to the Case Western Reserve experimental bench data. For different test bench data, what the results will be, it is recommended that the author should explain. Because the generalization of the model is very important, or the self-learning ability of the model is very important.  

Author Response

Dear reviewer:

I’m very grateful to your comments for the manuscript (Number: sensors-3139599, Title: Research on fault diagnosis of rolling bearing based on gramian angular field and lightweight model. Your comments are very important and helpful to my thesis writing and study. We have studied the valuable comments from you carefully, and tried our best to revise the manuscript. The manuscript has undergone English language editing by MDPI. The point to point responds to your comments are listed as following:

Point 1: To address issues such as the large number of parameters, extended runtime, and high memory usage associated with the ResNet-18 model, the article proposes a light weight fault diagnosis model named E-ResNet13. However, the author does not compare the research methods based on the current computing equipment, which is not convincing. 

Response 1: Thank you for your question. After the experiment in Section 4, we described the computing equipment used in the experiment and compared different diagnostic algorithms in 4.4.3, which were tested based on the equipment we described.

Point 2: For specific industrial application scenarios, do not know what kind of defects will occur in rolling bearings, and there is no pre-fault sample data of rolling bearing inner ring, outer ring, cage, and rolling element. Can your method still be applied ?

Response 2: Thank you for your question. What you have considered is correct. We have conducted sufficient experimental verification on the rolling bearing failure data set, which has not been verified in the actual industry. This is the work to be improved in this paper, and it is also the focus of subsequent research.

Point 3: The improved module 2 is applied in Fig.5, but it is not reflected in Fig.4.On the contrary, two improved modules 1 appear in Fig.4. 

Response 3: Thank you for your question. We changed (b) in Figure 4 to Improved module2 and added (a) (b) to the overall structure of the model in Figure 5.

Point 4: The paper uses the migration test to verify, but this is only limited to the Case Western Reserve experimental bench data. For different test bench data, what the results will be, it is recommended that the author should explain. Because the generalization of the model is very important, or the self-learning ability of the model is very important.  

Response 4: Thank you for your question. I quite agree with you that the generalization of the model is very important. Due to the excessive length of the paper and the time limit for the revision of the paper, I am worried that adding another data set will lead to too much length and more details of revision, which will further delay the submission of the manuscript. I implore modification on the basis of the proposed experiment. We are willing to continue making changes if necessary.

We tried our best to improve the manuscript and made some changes in the manuscript. We appreciate for reviewers’ warm work earnestly, and hope that the correction will meet with approval. Once again, thank you very much for your comments and suggestions. I hope I can have the opportunity to learn and communicate with you in the future.

Round 2

Reviewer 1 Report

Comments and Suggestions for Authors

This manuscript presents a Research on fault diagnosis of rolling bearing based on gramian angular field and lightweight model. Case studies demonstrated the effectiveness of the proposed method in detect fault. Some suggestions are below.

1. The summary of existing research in the introduction section of the manuscript is insufficient and needs further improvement.

2. The writing standards of the manuscript still need to be improved, and the reviewers have identified some obvious errors.

3. Some formulas in the manuscript have no references, so it is not a problem if all of these were proposed by the authors. However, if the formula is based on the work of others, corresponding literature should be provided.

4. The author only provided the analysis results of the proposed method in the experimental verification section, which is not enough. The author should provide comparative analysis results between the proposed method and advanced methods to highlight the effectiveness of the proposed method.

5. From Figure 13, the proposed method does not seem to have any outstanding advantages over DenseNet-121. In addition, since the proposed method is a lightweight model, it should be compared with a lightweight network, such as:

[1] A Lightweight Network with Adaptive Input and Adaptive Channel Pruning Strategy for Bearing Fault Diagnosis. IEEE Transactions on Instrumentation and Measurement, 2024, 73: 3510911.

Comments on the Quality of English Language

The overall quality of the manuscript is good, but there is still room for improvement in grammar and contrastive analysis.

Author Response

Please refer to the attachment for modification instructions

Reviewer 2 Report

Comments and Suggestions for Authors

Meet the conditions for publication, agree to publish

Author Response

Thank you for your approval of our manuscript, I wish you success in your work and happiness in your life